# One-Body Capillary Plasma Source for Plasma Accelerator Research at e-LABs

Sihyeon Lee [1], Seong-hoon Kwon [2], Inhyuk Nam [2], Myung-Hoon Cho [2], Dogeun Jang [2], Hyyong Suk [1,*] and Minseok Kim [2,*]

1 Department of Physics and Photon Science, Gwangju Institute of Science and Technology, Gwangju 61005, Republic of Korea
2 Pohang Accelerator Laboratory, Pohang 37673, Republic of Korea
* Correspondence: hysuk@gist.ac.kr (H.S.); kms83@postech.ac.kr (M.K.)

**Abstract:** We report on the development of a compact, gas-filled capillary plasma source for plasma accelerator applications. The one-body sapphire capillary was created through a diamond machining technique, which enabled a straightforward and efficient manufacturing process. The effectiveness of the capillary as a plasma acceleration source was investigated through laser wakefield acceleration experiments with a helium-filled gas cell, resulting in the production of stable electron beams of 200 MeV. Discharge capillary plasma was generated using a pulsed, high-voltage system for potential use as an active plasma lens. A peak current of 140 A, corresponding to a focusing gradient of 97 T/m, was observed at a voltage of 10 kV. These results demonstrate the potential utility of the developed capillary plasma source in plasma accelerator research using electron beams from a photocathode gun.

**Keywords:** capillary plasma source; plasma accelerator; e-LABs; capillary gas cell; computational fluid dynamics; discharge plasma; active plasma lens; particle-in-cell





## 1. Introduction

Plasma sources using gas targets have been extensively studied for their potential applications in laser–plasma acceleration (LPA) [1–7]. Capillary gas cells, which possess a small diameter along the longitudinal direction, have been commonly employed as a gas target due to their stability and controllability of the gas flow [5,8]. Early LPA experiments utilized gas cells as discharge plasma waveguides to enhance electron energy. The use of capillary discharge waveguides (CDW) featuring a parabolic density profile enables the transport of laser pulses over a prolonged distance beyond the Rayleigh length, resulting in an increased acceleration length and subsequently higher-energy electron beams [9–11]. In 2006, the group led by Wim Leemans at LBNL demonstrated a significant electron energy gain in LPA by using hydrogen-filled capillary discharge plasma [3]. They subsequently succeeded in achieving 8 GeV electron beams in 2019 [12]. In addition to their use as acceleration columns, discharge plasma sources have also been employed as plasma lenses for focusing electron beams [13–15]. Such plasma lenses, characterized by their high magnetic field gradient and low chromatic dependence, play a crucial role in LPA by enabling the production of a relatively large energy bandwidth in the electron beam. Unlike permanent magnet quadrupoles, active plasma lenses can produce azimuthally symmetric magnetic fields whose strength increases radially, enabling the symmetric focusing of electron beams.

The capillary cells were also demonstrated to be effective as steady-flow gas targets for LPA without the need for electrical discharge [5,16–19]. Upon irradiation with high-power laser pulses, gas breakdown occurs within the gas cell, resulting in the formation of plasma bubbles. Despite the absence of a preformed plasma channel, these cells have been shown to produce stable electron beams in terms of both reproducibility and beam quality due to the stationary nature of the gas distribution. In 2008, research conducted by Stefan

Karsch's group demonstrated the substantial advantage of utilizing capillary gas cells in generating reproducible, high-quality electron beams [5]. Subsequent studies verified the stable operation of plasma-based electron accelerators for prolonged periods of time, such as 24 h, utilizing the same concept [20].

Capillaries for LPA applications are typically constructed by combining two sapphire plates that have been laser-micromachined on their surfaces [19,21,22]. The sapphire material is known for its robustness against high temperatures, allowing it to withstand high-voltage discharged plasmas or tightly focused laser beams for prolonged periods of time. This property is particularly advantageous for LPA plasma sources that operate at high repetition rates or with high-intensity laser systems. However, there is a potential risk of gas leakage due to incomplete coupling of the sapphire plates. Alternative designs, such as one-body capillary cells made of materials such as glass or plastic, have been proposed to address this issue. However, the lifetime of these alternatives has not been fully established in LPA experiments [23,24]. Therefore, there is a need for the development of one-body capillary sources made from harder and more durable materials.

In this paper, we present the development of a sapphire-based, one-body capillary plasma source manufactured using a micromachining method with a diamond tool. The design and characteristics of the constructed capillary gas cell are outlined in Section 2. The performance of the gas cell is evaluated through laser wakefield acceleration experiments in Section 3. In Section 4, the potential application of the capillary discharge plasma as an active plasma lens is examined. Finally, we provide a summary and outlook in Section 5.

## 2. Development of the One-Body Capillary Plasma Source

### 2.1. Sapphire-Based, One-Body Capillary Gas Cell

Laser machining is a common method for the manufacture of sapphire-based capillary gas cells due to the material's hardness and the small diameter of less than 1 mm [9,21,22]. Short laser pulses are employed to engrave sapphire plates, creating gas feedlines and longitudinal capillary holes. In the case of cylindrical capillaries, semi-circular carving is implemented on each plate along the capillary axis, and the capillary gas cell is assembled by combining the plates. However, the machining conditions must be precise to fulfill the quality requirements of capillary shape and straightness [22]. In addition, an incomplete assembly of capillary cells may result in gas leaks and suboptimal capillary lines, particularly in longer capillary gas cells.

Diamond tool-based machining is an effective method for manufacturing harder materials such as sapphire [25–27]. In general, holes are produced by diamond drilling using an electroplated diamond tool, in which diamond grains are electrodeposited onto metal or alloy substrates. As illustrated in Figure 1, a sapphire capillary cell was manufactured using diamond machining. All processing to create the capillary gas cell was performed using diamond drill bits on a single block of sapphire by a local company (COMA Technology Co., Gumi, Republic of Korea). The capillary and gas lines of a 1 mm or larger hole diameter were formed using the diamond drilling. The hole depth depends on the tool design of the manufacturers and, in this paper, the longest capillary produced was 15 mm in length with a 1 mm diameter. This length was achieved by drilling from one side of a sapphire block; thus, it can be extended to 30 mm by drilling on both sides. Capillaries with smaller diameters can be manufactured using thin, standard diamond drill bits. Although the restricted length leads to a limitation of the hole depth, it is possible to extend the capillary length through the coupling of multiple sapphire blocks. The design of capillaries for the stated objective is currently in progress and will not be covered within the scope of this paper. In addition to hole processing, internal thread manufacturing for gas delivery and the capillary holder was adopted. One-touch fittings (EOCC03M3, KCC Co., Ltd., Seoul, Republic of Korea) were connected to the sapphire block for gas inlets. The simple design of this one-body gas cell has allowed for the realization of a compact capillary gas cell without the need for housing, which is typically required in sandwiched capillary designs. The one-body design of the gas cell allows for a compact and leak-free capillary gas cell that

does not require housing, making it easily adaptable to various plasma accelerator research. Furthermore, for discharge plasma applications, oxygen-free electrolytic copper electrodes were installed at both ends of the gas cell, and the utilization of PEEK material for the bolts and holders close to the electrodes has prevented unexpected discharges. Discharge experiments using electrodes will be discussed in Section 4.

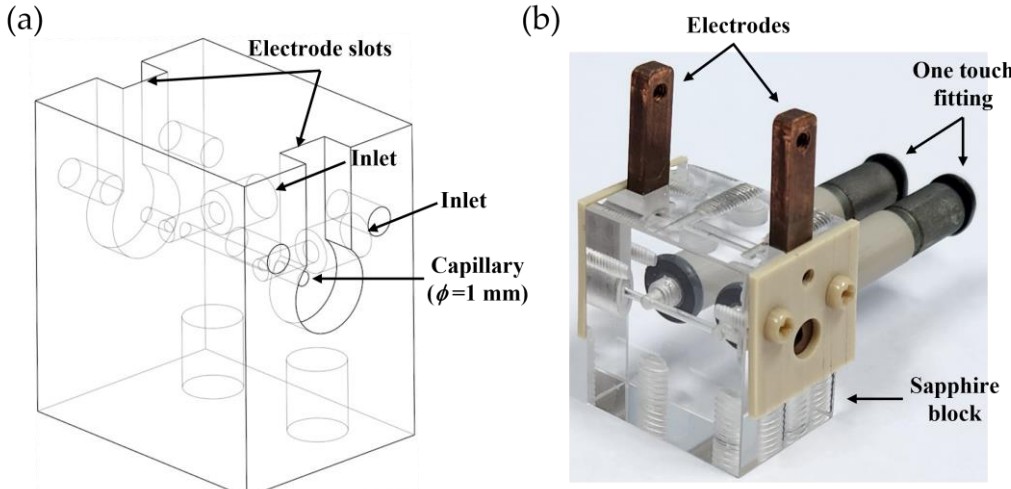

**Figure 1.** (**a**) Designed and (**b**) manufactured one-body capillary gas cell made of a sapphire. For the gas delivery, two one-touch fittings are directly connected to the sapphire block. Electrodes are attached at each end of the capillary and the tapped holes at the bottom connect the holder.

### 2.2. Characterization

The effective utilization of the capillary plasma source requires a comprehensive understanding of the gas density distribution within the capillary. To characterize the capillary density profile, we employed Mach–Zehnder interferometry and computational fluid dynamics (CFD) simulations with a 1 mm diameter capillary of a 7 mm length. The experimental setup for the interferometry is represented in Figure 2a. The capillary system was positioned in a vacuum chamber operated at $1.2 \times 10^{-4}$ Torr. The interference images were acquired using a continuous wave He-Ne laser (632 nm wavelength) and a camera (Marlin-F-131B) capture rate of 67.27 frames per second. The laser was separated into two beams (probe and reference) by the first beam splitter. The laser beam was expanded to a size larger than the sapphire block, enabling the acquisition of a reference image of its side. The signal image was obtained via the probe beam passing through the capillary gas cell, while the reference image was obtained by the probe beam bypassing the capillary, effectively reducing the noise induced by the vacuum pump system imprinted on the reference image. The second beam splitter recombined the two beams, and the mirrors positioned after the splitter reflected both the signal and reference images of the probe beam. This setup is designed to yield both a magnified signal and reference images separated spatially in a single image, as is depicted in Figure 2b.

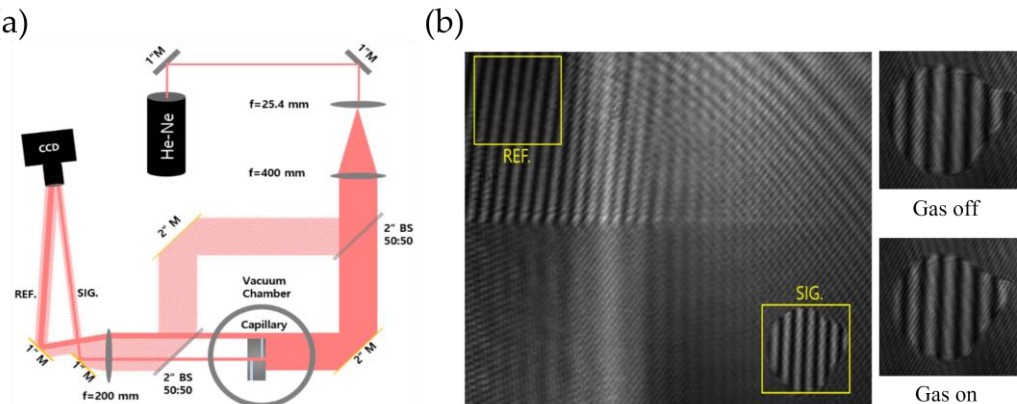

**Figure 2.** (**a**) Experimental setup for the gas density measurement using a Mach–Zehnder interferometer. To obtain signal and reference images in the single shot image, two mirrors were used for the separate adjustment of the image position. (**b**) The captured image shows the magnified interference patterns, including both signal and reference images.

The gas pressure inside the capillary cell was determined by analyzing the phase difference between two signal images obtained before and after the gas injection. A reference image, shown in Figure 2b, was used to eliminate the noise data, which can be induced by vibration. The resulting phase shift was achieved using a fast Fourier transform (FFT) method. The relation between the gas pressure ($P$) and phase shift is expressed as $P = (2RT\Delta\phi)/(3Ak_0L)$, where $\Delta\phi$ is the total phase shift accumulated along the gas density, $A$ is the molar refractivity, $R$ is the gas constant, $T$ is the temperature in Kelvin, $k_0$ is the wavenumber of the laser, and $L$ is the gas length [28]. In the measurement, nitrogen gas was utilized with a molar refractivity of 4.459 $cm^3$/mol [29], and the pressure was varied from 200 to 500 mbar while maintaining a constant gas opening time of 300 ms. In the longitudinal interferometry, a problem arises in specifying the gas length, $L$. Since the pressure is dependent on the longitudinal position, $x$, the total phase shift is a function of both pressure and the length.

$$L_{eff} = \int_{-\infty}^{\infty} P(x)dx / P_{\max}. \tag{1}$$

where $L_{eff}$ is the effective length and $P_{\max}$ is the maximum pressure. Thus, the position-dependent pressure distribution is required to determine the total phase.

To achieve the phase, a three-dimensional (3D) CFD simulation was conducted using ANSYS FLUENT software (version 2019-R1). The simulation conditions involved continuous gas injection at a constant pressure from the inlets and outlets (the exit of capillary) in a vacuum state. Figure 3a shows the CFD simulation results of the obtained gas density distribution. In this example, the capillary length was 7 mm and the effective length was calculated to be 5.69 mm using Equation (1). Using these values, the gas density was calculated as a function of time (Figure 3b). The maximum gas density was reached after about 50 ms and was maintained for the duration of the gas opening time (300 ms). These results suggest that our capillary cell provides a stable gas distribution.

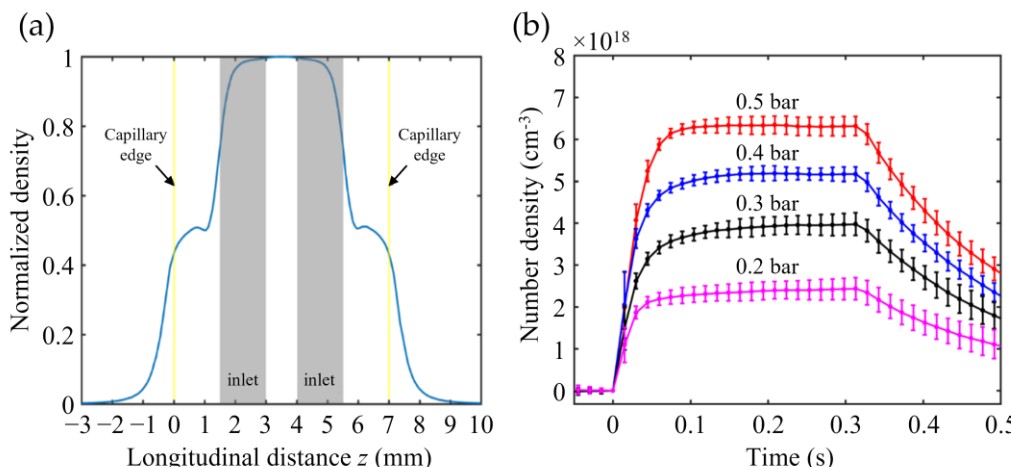

**Figure 3.** (**a**) CFD simulation result for the 7 mm long capillary gas cell. (**b**) Gas density temporal evolution in the capillary gas cell. The gas density was stabilized after about 50 ms of gas-filling time.

### 3. Laser–Plasma Electron Acceleration with a Capillary Gas Cell

To investigate the capabilities of the one-body capillary gas cell for electron acceleration in LPA, we conducted the electron acceleration experiments using a 150 TW laser system at the Institute for Basic Science (IBS), GIST [30]. A schematic of the experimental setup is illustrated in Figure 4. The laser used in the experiment had a center wavelength of 800 nm and a bandwidth of 76 nm (FWHM), and the measured laser pulse duration was 25 fs (FWHM). The laser beam reflected by a holed mirror was focused using a spherical mirror with a 1 m focal length, resulting in a laser spot size of 24 μm (FWHM) at the focus. The normalized vector potential ($a_0$) of approximately 1.6 was determined by considering the encircled energy fraction in the spot size, the reflectivity of the deformable mirror, and the reflectivity of the holed mirror. A 7 mm long, one-body capillary gas cell was mounted on a 5 axis motorized stage inside a vacuum chamber (~$4 \times 10^{-5}$ Torr) and was used for the laser wakefield acceleration (LWFA) experiment. The cell was filled with pure helium gas at a pressure of 150 mbar. Considering the full ionization of helium gas (ionization intensity of $8.8 \times 10^{15}$ W/cm$^2$) by the intense laser pulse used in this experiment (the peak laser pulse intensity of ~$10^{19}$ W/cm$^2$), the estimated plasma density was $7.2 \times 10^{18}$ cm$^{-3}$. The laser was deflected by a tilted aluminum foil after passing through the capillary, and the electrons generated were diagnosed as they passed through it. The diagnostic devices were covered by black-anodized foils to prevent stray light from the laser-induced plasma and deflected laser pulse from reaching them. To analyze the energy spectrum of the electron beams, two phosphor imaging plates (LANEX 1, 2) were placed before and after a 20 cm long permanent dipole magnet with a field strength of 1 T. The trajectory deviation recorded in the images captured by LANEX 1 was used to calibrate the electron energy spectrum recorded at LANEX 2.

The electron beam spectra obtained from six consecutive shots are shown in Figure 5a. The beam energy, which remained nearly constant at $200 \pm 25$ MeV, highlighted the stability of the newly developed capillary gas cell as a plasma acceleration source. Figure 5b provides a comparison with the particle-in-cell (PIC) simulation results, which were performed using the cylindrical grid algorithm, Fourier-expanded in the azimuthal direction [31]. The simulation parameters were similar to the experimental conditions, with a normalized vector potential of the laser $a_0 \approx 1.6$, laser spot size of 25 μm, pulse duration of 25 fs, and plasma density of $n_0 = 7 \times 10^{18}$ cm$^{-3}$. The simulation box dimensions were $z \times r = 60$ μm $\times 200$ μm, $\Delta z = \lambda_{laser}/25$, and $\Delta r = 10\Delta z$, with macroparticles of $N_z \times N_r = 2 \times 6$ and azimuthal modes of 0, 1, and 2. The overall plasma profile was consistent with the CFD simulation (Figure 5c), and the laser focus was targeted at $z = 3$ mm, which was 1/3 of the way into the flattop region. Results showed that the maximum electron beams were generated at a plasma position of approximately $z = 3.5$ mm (Figure 5d).

After this point, the laser peak intensity began to decrease, accompanying weaken wakefields. As the laser propagated, the laser-driven wakefields disappeared and the injected electron beams started to create beam-driven wakefields, which then generated and accelerated the secondary electron beam, as shown in Figure 5e. While the first electron beam lost some of its energy exciting wakefields, causing its energy spectrum to broaden, the second electron beam displayed a narrower energy spectrum (Figure 5f) that was comparable to the experimental results in Figure 5b. The first electron beam reached a high energy of approximately 800 MeV; however, it was not detected in the experiment as a result of the limited measurement capability of the electron spectrometer at 400 MeV and the low charge of the electron beam. We note that since this experiment was conducted to demonstrate the potential of the developed capillary gas cell as an accelerating column, the conditions were not optimized during the experiment. According to the simulation results, it could be inferred that better electron beams may be obtained by either increasing the laser intensity or using a shorter capillary cell of approximately a 4 mm length.

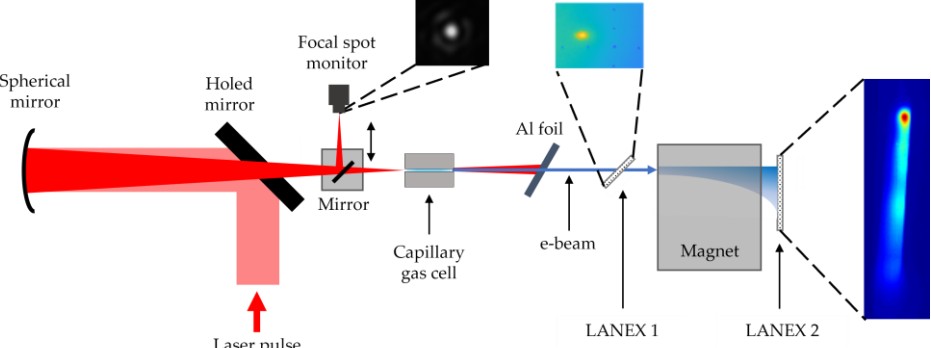

**Figure 4.** Experimental setup for a laser wakefield electron acceleration. The 150 TW laser system and a helium-filled capillary gas cell of 7 mm length are used.

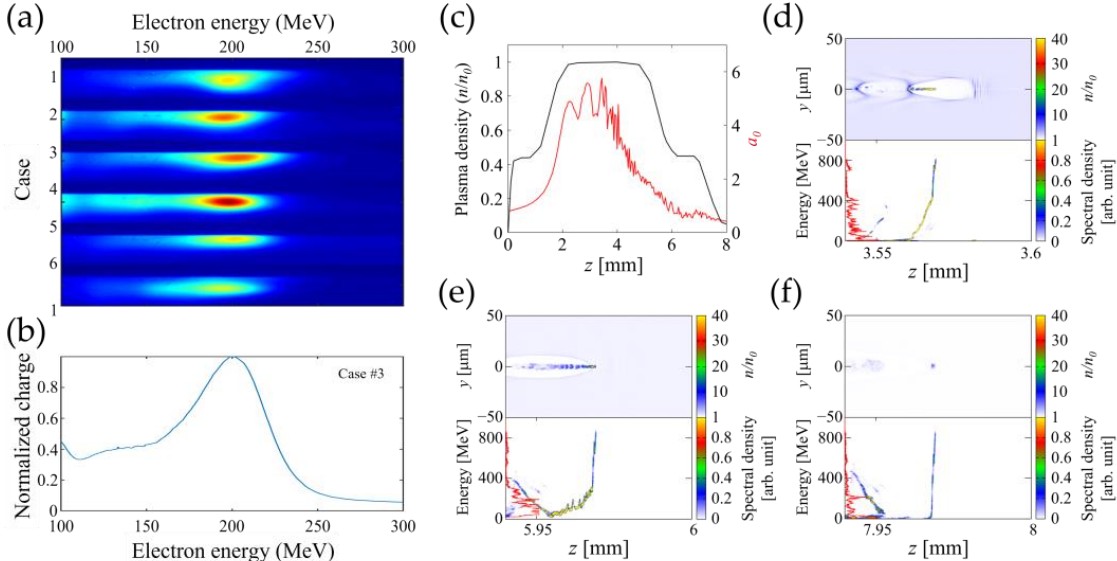

**Figure 5.** (**a**) False color images of the accelerated electrons on the LANEX 2 and (**b**) normalized charge as a function of electron energy. (**c**) The plasma profile was set similarly to the CFD simulation result. The PIC simulation results at three different positions are represented in (**d**–**f**). The red lines in left of the bottom panels are the spectral density in terms of beam energy.

## 4. Capillary Discharge Plasma Source for Active Plasma Lens

In this study, we investigated the feasibility of utilizing a capillary gas cell as a source for a plasma lens. The plasma lens is an attractive method of focusing electron beams

through the creation of a radially symmetric field, as reported in previous studies [13–15]. Our approach involved the use of a pulsed, high-voltage (HV) discharge system to generate a gas discharge plasma, as is depicted in Figure 6a.

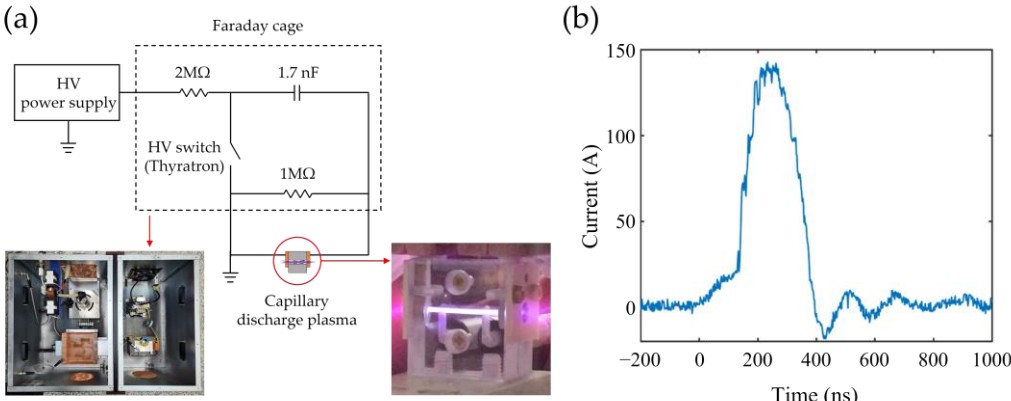

**Figure 6.** (**a**) Schematic diagram of the pulsed, high-voltage discharge system. The pulsed, HV discharge system and a 15 mm, helium-filled capillary discharge plasma are shown in the inner photographs. (**b**) The current profile, with a peak current of 140 A, was obtained by applying the voltage of 10 kV.

The HV discharge was triggered by a thyratron switch (MA2440B, E2V) and powered by a DC power supply (EH50R02, Glassman) capable of providing up to 50 kV. All circuit components, depicted by the dashed line in Figure 6a, were installed in a metal box for radiation shielding. The gas pressure and injection were controlled by a vacuum regulator (3491A, Matheson, Irving, TX, USA) and a solenoid valve (6013A, Burkert, Ingelfingen, Germany), respectively. The capillary cell employed was 15 mm long, with a diameter of 1 mm, and was fitted with holey copper electrodes, which were mounted at each end with PEEK cover plates, as shown in Figure 1b. Helium gas was introduced into the capillary at a constant pressure of 300 mbar and a high voltage of 10 kV. The operation and synchronization of the gas injection and high voltage systems were managed by a delay generator (DG645, SRS). The generated capillary discharge plasma is shown in Figure 6a. The current was monitored during the discharge using a current transformer (7795, Pearson Electronics, Palo Alto, CA, USA), and is depicted in Figure 6b. The current increased to 140 A after an initial breakdown phase of approximately 100 ns.

Assuming a cylindrical coordinate, the distribution of the magnetic field corresponding to the measured current can be calculated using the Ampere's law, as follows:

$$B_\theta(r) = \frac{\mu_0}{r} \int_0^r J(r')r'dr', \tag{2}$$

where $B_\theta$ is the azimuthal magnetic field, $\mu_0$ is the magnetic permeability of vacuum, and $J(r)$ is the current density within $r < R$ ($R$ is the capillary radius). The current density, $J(r)$, is a varying function along the radius, $r$, which can be derived by the assumption of a plasma temperature difference between the center ($r = 0$) and the capillary wall ($r = R$) [32]. It is anticipated that the $B_\theta$ exhibits a non-linear profile as a result of the varying current density. Figure 7a shows the current density and the azimuthal magnetic field along the radius at the discharge current of 140 A in a 1 mm diameter capillary. The magnetic field gradient around the center was determined to be 97 T/m. To validate the beam focusing capabilities in the discharge capillary, the PIC simulations were performed using Equation (2), and the numerical, dispersion-free field-solver was employed to minimize the numerical Cherenkov noise [31]. The simulation range was at a capillary length of 15 mm, and the beam size calculation was then followed using the Twiss matrix. The Twiss parameters for an input electron beam were $\beta_r = 10$ m; $\alpha_r = 7$; a normalized emittance, $\epsilon_{Nr} = 0.5$ µm mrad; the

beam energy, 70 MeV; and a root mean square (rms) beam size of $\sigma_{r,rms} = 179$ μm. These are from the beam parameters provided at the photocathode, gun-based electron accelerator facility (called e-LABs) of the Pohang Accelerator Laboratory (PAL) [33,34]. An increase in the discharge current resulted in a reduction in the focal distance; however, but the focused beam size remained constant (Figure 7b). On the other hand, a higher current increased the beam emittance due to the larger non-linearity of the magnetic field profile near the capillary wall. Thus, although realizing a high focusing magnetic field of kT/m in a capillary with a diameter of 1 mm, which is relatively large, presents a challenge, there is a benefit in establishing a region of linear magnetic field (approximately half of the capillary diameter) [13] that contributes to the preservation of beam emittance.

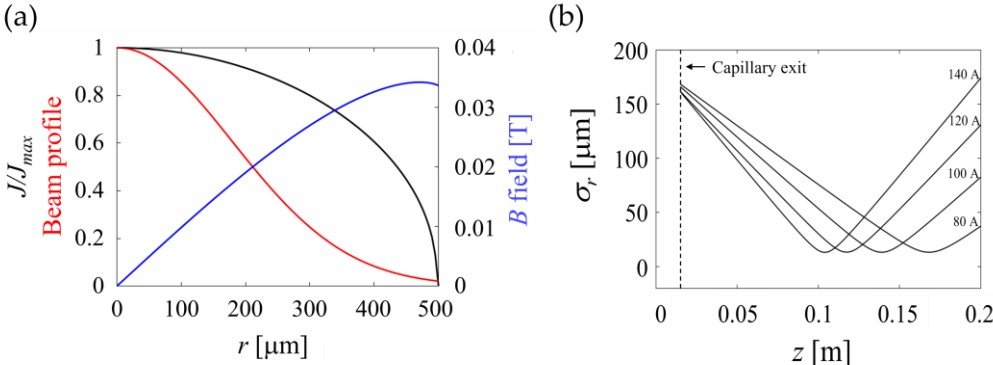

**Figure 7.** (**a**) Normalized current density and the azimuthal magnetic field in a 1 mm diameter capillary plasma. An electron beam with a rms size of 179 μm was used to investigate the focusing property by the developed discharge capillary. (**b**) Different focal position in terms of the discharge current. The spot size was maintained at 13 μm regardless of the current.

## 5. Discussion

We successfully developed and investigated a sapphire-based, one-body capillary gas cell for plasma accelerator applications. The sapphire capillary gas cell was manufactured by diamond machining. Due to its simple, one-body structure, there was no need to be concerned about gas leakage and assembly. The operational properties of the gas cell were investigated by using the longitudinal interferometry and CFD simulations. In gas density measurements, the result showed that the maximum gas density was reached after 50 ms of gas injection. To confirm the stability of the capillary gas cell, a laser–plasma electron acceleration experiment with the capillary cell was performed using a TW laser system. From the LWFA experiment, a stable and reproducible electron beam of 200 MeV was generated. The experimental results were analyzed using the PIC simulation, and it was demonstrated that both results were in good agreement. In addition, to investigate its potential application as an active plasma lens, a pulsed discharge system was applied to the gas cell to generate a capillary discharge plasma. An analysis of the discharge current of 140 A predicted the capability to achieve a focusing gradient of 97 T/m. Although it is necessary to find more optimal conditions according to the property of electron beams generated at e-LABs, the possibility of its use as an active plasma lens was demonstrated. The characteristic results of the capillary plasma source shown in this paper provided useful information for its application in LPA, with external injection planned to be carried out in the e-LABs facility (e-LABs is an acronym standing for "electron linear accelerator for basic science").

**Author Contributions:** Conceptualization, I.N., H.S. and M.K.; software, S.L. and M.-H.C.; formal analysis, S.-h.K. and D.J.; investigation, S.L., S.-h.K., I.N., M.-H.C. and M.K.; writing—original draft preparation, S.L. and M.K.; writing—review and editing, M.-H.C., D.J., I.N., H.S. and M.K. All authors have read and agreed to the published version of the manuscript.

**Funding:** This research was supported by the National Research Foundation of Korea (NRF) funded by the Korea government (MSIT) [No. 2020R1C1C1011840, 2020R1C1C1010477, 2021R1C1C1003255, 2021R1F1A1062911, 2022R1A2C2009768, 2022R1A2C3013359].

**Institutional Review Board Statement:** Not applicable.

**Informed Consent Statement:** Not applicable.

**Data Availability Statement:** The data presented in the study are available on request from the corresponding author.

**Acknowledgments:** The authors acknowledge all e-LABs and CoReLS members who participated in operations.

**Conflicts of Interest:** The authors declare no conflict of interest.

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
