# Peer review of "One-Body Capillary Plasma Source for Plasma Accelerator Research at e-LABs"

_applsci, doi:10.3390/app13042564_

Round 1
Reviewer 1 Report
The manuscript basically states that the authors built a capillary discharge, shot a short laser pulse through it and observed electrons emerging from the system. It shows that the authors can perform, and have the equipment suitable for such experiments and for an active plasma lens experiment. None of the results are particularly special or interesting. The authors claim good agreement between experimental and simulation results, though no more than two normalized energy spectra are shown.
The real laser pulse duration (measured or inferred, the real laser focus shape, the timing in the discharge, and many other key parameters are not given. Considering that, it may be surprising that such a good agreement is found between experimental and simulation results.
Both the description and the experiment and the simulations and their results are presented at a very superficial level. On the one hand, the results presented "make sense", I.e., there is nothing that allows the referee to dismiss them. On the other hand, it is difficult to validate them. For example, focusing gradient is estimated from Eq.2, but no information is given on the shape of the current density profile (assume constant density?).
It makes sense that with this discharge radius, there is no laser guiding and the a0 value of the laser pulse will quickly decrease along the plasma. Whether LWFA and then PWFA occur may be true, but nothing is shown.
It is not clear what experiments that has not been performed before the authors plan on performing with such a system. External injection in a plasma-based accelerator would require some sort of well-contolled, constant gradient, free of dark current, etc. plasma-based accelerator and maybe one or two plasma lenses. It is not clear to me that what is presented in the manuscript demonstrates that any of that can be done.
The decision is thus in the hands of the authors: do they want such a vague manuscript to be published with their good name as authors?
My recommendation is that the authors give a clear (as opposed to a vague one) motivation for why they present the results in the manuscript. Then they should point out why what is shown demonstrates that what they want to do can be done. For example, does the single-body discharge satisfy any or the requirements or is that just another plasma discharge source? Once that done, the manuscript could be considered for publication.
Reviewer 2 Report
The manuscript "One-body capillary plasma source for plasma accelerator research at e-LABs" by Sihyeon Lee et al. provides a detailed description of gas-plasma devices for the planned experiments in e-LABS. Although the technical features of the proposed devices are original enough, in order to determine the degree of their applicability in experiments in other laser laboratories, the description of the devices should be extended.
In particular, it is not entirely clear from the manuscript whether the authors are going to use identical gas systems as an active plasma lens and as an electron acceleration system? Also, HV electrodes are not shown in the figures.
When calibrating the gas density, the authors show the interferograms, but do not provide a detailed description of their processing. How do the authors calculate the phase difference between signal and reference images? Could the authors provide interferometric images for two different pressures in the gas system?
Can a gas system with an internal hole diameter of 1 mm be considered a capillary? And does a tube of this diameter affect the acceleration process? Would it not turn out that the electrons would have reached a similar energy if the authors had simply used an ordinary gas cell or a gas jet?
Reviewer 3 Report
The authors report the development of a capillary plasma discharge device similar to a few others except that it is made by some drilling technique that allows it to be made from a single block of sapphire instead of joining two.
My recommendation is that this paper should be resubmitted after the authors clarify the following points…
The authors claim that a single body allows the reduction of gas leaks. It seems a weak reason to to justify a different technique since moderate leaks can in general be tolerated in a system that have large leaks by design (the apertures of the capillary).
The authors don’t discuss in detail other advantages that may result from the manufacturing technique.
The manufacture from two slabs have an important potential advantage ver the single body: the possibility of making long capillaries (several cm at least). The authors do not discuss what are the maximum capillary lengths that can be achieved by this technique.
Capillary devices using electrical discharges where developed for LWFA’s aiming to use its plasma density radial profile that can be temporarily close to parabolic (with a minimum on axis) and therefore have a focusing effect on the laser. The authors do not mention this aspect of the device. In such a short capillary, and in the contexts of a LWFA, the main reason to use a discharge is probably to obtain this focusing effect, otherwise the LWFA may perform simillarly without the discharge, using it as a neutral gas cell. Of course this is not the case in the context of a charge particle beam plasma lens since it is the discharge current that makes the Magnetic field. The authors should address this aspect and if the focusing capability is to be used appropriate references should be added.
Due to the manufacturing technique the capillary inner diameter is, according to the authors 1 mm. This diameter seems to large for a plasma lens since the current density do not reach a high value. The authors do not discuss this problem.
The authors should provide more detail on the drilling technique used, since this is a core aspect of the paper.
I am guessing that the measurements of gas neutral density presented in figure 4-b are the result of the CW interferometry process, that should be clearly sated in the paper.
The authors suggest (lines 182-184) that laser ablation in a capillary can lead to plasma lensing (acting on charged particles), in my view the plasma lensing effect is due to the magnetic field radial distribution (according to the paper equation 2). Can the authors explain how laser ablation lead to plasma lensing in a capillary?
The authors estimate the magnetic field (line 204) but they do not explain how they made the estimation. In my view the authors assumed a radial distribution (J(r’) as in equation 2) to make that estimation, the authors should provide more information on this point.
Line 218 substitute “hole mirror” by “holed mirror” or mirror with a hole” or similar.
In the discussion the authors do not discuss why they need to use the discharge or what is the effect of the discharge in the LWFA performance, what would be the effect of using a twice the voltage, or increase the circuit capacitor, etc.
The paper should receive a grammar revision before resubmission.
